# Analysis of a Historical Accident in a Spanish Coal Mine

**DOI:** 10.3390/ijerph16193615

**Published:** 2019-09-26

**Authors:** Lluís Sanmiquel-Pera, Marc Bascompta, Hernán Francisco Anticoi

**Affiliations:** 1ICL Chair in Sustainable Mining, 08242 Manresa, Spain; marc.bascompta@upc.edu; 2Department of Mining Engineering, Industrial and ICT, 08242 Manresa, Spain; hernan.anticoi@upc.edu

**Keywords:** coal mining, ventilation, accident analysis, explosion

## Abstract

There has been a long history of coal mine accidents and these, usually, involve serious injuries, fatalities, and the destruction of facilities. In the seventies, an explosion killed 28 miners in a Spanish coal mine. This paper gives insight into the main factors of the accident by means of the causation mode, using two well-known alternatives: (1) the method from the Spanish Instituto Nacional de Seguridad y Salud en el Trabajo (INSST), where the causes and circumstances of the accident are classified into immediate causes and basic causes, and (2) the Feyer and Williamson method, where the classification is done using precursor events and contributing factors. The analysis identifies the lessons to be learned from the disaster. Both methods have given very similar results, verifying the goodness of the analysis. Methane emissions due to a variation in the exploitation method, the electrical installation, and a lack of safety procedures and training were the main causes of the accident. These findings explain the real causes of this accident and can be very valuable for the prevention of future accidents.

## 1. Introduction

Safety issues are a subject of great concern in the mining industry [1,2,3,4], with proposals of new approaches to achieve better explanations and predict and prevent workplace accidents [5,6]. Coal mining is one of the most dangerous types of mining within the sector [7,8], and many studies have been done to understand the main causes of hazards such as coal dust explosiveness or methane [9,10,11,12,13,14,15].

Coal mining has suffered many fatalities over the last century despite the obvious advances in technology [16,17]. The vast majority of this type of accidents are related to explosions and deflagrations due to poor ventilation and inadequate conditions or exploitation method. Mine explosions, whether of methane or coal dust, and spontaneous methane emissions are the most common causes of high fatalities in coal mining accidents [18,19,20,21]. Besides, several studies have revealed that some factors such as environmental conditions with a significant presence of dust, humidity, and falling rocks can act as precursor agents to accidents in underground mining, increasing the incidence and severity of accidents [22,23,24,25,26]. Hence, it is important for the stakeholders involved in the industry to have the proper tools, frameworks, and understanding of the hazards to deal with accident causation and to achieve a proper safety system [7,27].

As in other historical accidents that have been restudied after many years [28,29], this research is focused on analyzing an explosion that occurred in a coal mine that caused the death of 28 miners. Initial information suggested the accident suffered in the case study was produced because of an inadequate exploitation method, but some points still remained unclear. For this reason, we have used two analysis methods to explain and verify the events, which were unknown when the accident occurred [30,31,32,33]. Interviews have also been carried out in order to obtain additional information. The aim of the study is to clarify the main factors of the accident by means of two well-known methodologies, as well as highlight the main differences and similarities between them.

## 2. Case Study 

### 2.1. Exploitation Method 

The mine seams were between 11° and 15° north and mining was taking place in depths between 525 and 707 meters. The exploitation method used at that moment was a classic longwall system, with panels of approximately 250 meters width, a slope between 12°–15° and two main drifts with a difference in level of around 50 meters. A mechanical shearer with a self-advancing hydraulically powered supports was implemented about one month before the accident, changing from the old support and extraction method, manual drilling, and wooden frames. A conveyor belt system was used to carry out the coal through a ramp. The mine was equipped with two main fans and then the airflow was conducted to the different workshops by means of forcing auxiliary fans. Initially, the changes in the exploitation method were thought to be the only responsible of the accident.

### 2.2. Methane Characteristics

An atmosphere containing between 5% and 15% methane and over 12% oxygen can be an explosive mixture. The temperature required to ignite an explosive methane-air mixture is approximately 540 °C, being easily ignited by an electrical arc, frictional spark, heated surface, or open flame. The amount of energy necessary for ignition will vary with gas concentration; however, as little as 0.3 millijoules (mJ) of electrical energy is required. Roof falls can ignite explosive methane–air mixtures either by generating frictional heat or by releasing piezoelectric energy [16].

Besides, high concentrations of gas in the goaf and adjacent layers can pour into the workings as the air pressure in different parts of the underground mine is imbalanced [34]. Despite the fact that the gas emission zone can be estimated globally using various methods or assumptions, it is not uncommon that predictions may be under or overestimated due to the lack of sufficient spatial information defining the quantity and location of the gas sources in the overlying strata [35].

According to Spanish legislation, in the moment of the accident, methane emission measures done in the panel of the accident were considered as very low, 1.4 m^3^/ton. However, methane measures were taken three days after the accident without ventilation and concentrations of 5% of methane were measured, which was the maximum level of measurement for the type of methanometer used. 

Originally, it was thought that the source of such amount of methane was some fractures in the coal seam. Subsequently, a set of measures was done to determine its behavior. Table 1 gathers the measures done in the most adverse points of the roof after the accident, Point A and Point B from Figure 1, switching off the ventilation system.

Values from Table 1 exposes that: (a) Methane concentration was very low within the normal functioning of the auxiliary fan, and (b) there was a constant increase of methane concentration once the fan was switched off until high values were above the capacity of the methanometer. The values suggest that methane ascends to the inlet due to very low natural ventilation, in the opposite direction of the artificial ventilation.

### 2.3. Accident Details

The accident occurred due to a very strong explosion. The first shift entered to the mine at 8:00 am and the accident was at 9:00 am. 28 miners died in the *Working Face 09*, Figure 1, where number 1 is the entry of the workshop and number 2 is the exit. This workshop had a length of 224 meters with a seam between 1.5–1.8 meters and a slope of around 11°. Figure 2 shows the characteristics details of the workshop.

The inspection of the accident concluded that it started in the first third of the panel and the expansion was in both directions, to *Drift-09* and *Cementos-07*. However, the shock wave was diluted in the lower part of the panel, *Drift-09*. The medical survey pointed out that the death had been immediate due to severe burn injuries with a low quantity of toxic gases or mechanical effects.

## 3. Methodology

The causation models are often used to explain accidents, either in the mining industry or in other sectors [36,37]. The cause diagram analysis is a widespread system to determine the basic and immediate causes of an accident or incident by means of simply organizing the factors collected in situ. This paper analyses the causes and explains the origin of the accident. Government reports and interviews with the technical staff, miners, and doctors have been used to support it, as well as measurements done in the mine after the accident. The method chosen in this case study has been used in many other similar investigations, such as for [29,30], because of its appropriateness in considering the sequence of events leading to the accident.

The accident investigation is based on the Instituto Nacional de Seguridad y Salud en el Trabajo (INSST) method [30,31] and the Feyer and Williamson method [32,33].

The INSST method exposes a classification of the two different causes: Immediate and basic. Immediate causes are the circumstances that occur just before the accident and they are usually divided in unsafe acts (UA) and unsafe conditions (UC), depending on whether the indicated circumstances are due to the behavior of the workers or workplace bad conditions. Therefore, unsafe acts are mainly attributable to workers, while unsafe conditions are usually attributable to managers, work equipment, or lack of procedures, among others. The basic causes are the reason because unsafe acts and conditions happen and they are usually organizational failures that facilitate immediate cause such as technical and behavioral failures. These basic causes can be classified in personal factors (PF) or workplace factors (WF), which cause unsafe acts and unsafe conditions, respectively.

On the other hand, the Feyer and Williamson method (Figure 3) allow determining up to three events that preceded the accident, called precedent events, and are crucial for the accident occurrence. These are called Precursor Events (abbreviated to PE1, PE2, and PE3). Each of these events could be classified in one of four possible ways:
E1. Environmental events (EE): Events or conditions resulting from the location of the accident (e.g., low lighting, wet floor, or cramped conditions).E2. Equipment events (EQ): Events resulting from breakage or malfunction of machinery or tools.E3. Behavioural events (BE): Events resulting directly from human involvement (e.g., leaning too far into the path of machinery, touching an electrically charged object).E4. Medical events (ME): Events resulting from the person’s current state of physical well-being (e.g., heart attack or diabetic or epileptic episode).

Behavioral events, E3, are coded further into whether they constitute an error or not (i.e., the incorrect performance of standard operating procedure). Errors are sorted in two well-known classification systems. The first coded error into omissions (things not done) or commissions (things done incorrectly). The second one into skill-based errors (routine behavior), rule-based errors (involving the application of learned rules) and knowledge-based errors (troubleshooting).

Besides, the Feyer and Williamson method allows identifying contributing factors that were crucial for the sequence of the accident consisting of 1, 2, or 3 events. If any of the events did not happen, the accident would not occur. However, the contributing factors alone are not enough for the accident occurrence. It is also important to note that the contributing factors largely coincide with the basic or main causes of the INSST method.

The nature of contributing factors (CF) was also coded into eight possible categories:CF1. Environmental (E): Factors resulting from conditions occurring at an earlier time at the location of the accident.CF2. Equipment (EQ): Factors associated with the design or upkeep of machinery, tools, personal protective equipment, or safety equipment.CF3. Work practice (WP): Factors involving risky standard operating procedures accepted by management and/or personnel (including categories of poor upkeep or misuse of equipment).CF4. Supervision (S): Factors relating to inadequate charge of workers.CF5. Training (T): Factors relating to inadequate training of workers.CF6. Task error (TE): Factors relating to incorrect performance of duty.CF7. Medical (M): Factors involving physical well-being at an earlier time.CF8. Other (O): Factors such as alcohol/drug involvement, delays in reaching medical attention, and social aspects.

## 4. Results

### 4.1. INSST Method

The nomenclature followed is described in the INSST method [30]. IC means immediate cause; BC means basic cause; *Fxx* means type of fatality; *Dxx* deviation; *UAxx* unsafe act; *UCxx* unsafe condition; *PFxx* personal factor, and *WFxx* work factor. All these elements are included in the INSST method [30,31]. 

#### 4.1.1. Immediate Causes

A conjunction of two unsafe conditions and one unsafe act were identified in the root of the accident:-UC1: Inadequate protection or safety guard in the electrical equipment for an explosive atmosphere at the moment it was switched on. The data collected after the accident suggested that a spark in the electrical light system of the head drift got in contact with a coal bed methane due to the lack of electrical protection against explosions. The event happened before the ventilation system was able to dilute the methane concentration in the roof the drift.-UA3: Incorrect control, done by the watchman at the beginning of the shift after a production break. There is no record of controlling the working faces before the shift restarted working and switched on all the equipment.-UC6: Danger of fire or explosion due to an explosive mixture in the drift.

#### 4.1.2. Basic Causes

-The unsafe act UA3, previously stated, was produced because of a lack of knowledge/training of the watchman, who did not know that a safety protocol to find coal bed methane in the drifts had to be applied before the entry of the workforce and initiation of the installation. The mining company did not provide any procedure, documentation, or training in this regard. This fact is a basic cause in the second level of the event sequence, lack of knowledge of the employee (PF2).-A combination of two basic causes, also in the second level of the event sequence, resulted in an unsafe condition in the first level (UC1):
○WF1: Insufficient leading and supervision, due to the mine not being properly classified with regard to the danger of gas emissions.○WF4: Inadequate maintenance of the installation and/or equipment due to a lack of control of the electrical equipment, which was not appropriate for a coal mine.-Another combination of two basic causes in the second level of the event sequence gave a first level of unsafe condition (UC6):○WF6: Poor work procedures. The company did not have any procedure before switching on the installation once the ventilation system had been stopped.○WF1: Leadership and lack of supervision. The staff of the mine did not take into account the increase of the methane emission caused by the change of the production method 23 days before the accident. More productivity gave more release of methane.

Figure 4 summarizes the accident by means of a causal tree using the INSST method. The specific conditions of the accident are also detailed below.

-F13: Type of accident by contact with direct flames or objects/environments with high temperature or in flames.-F41: Type of accident by collision or blow against an object or fragments—projected.-D13: Deviation or anomalous event due to an explosion.-UA3: Unsafe act due to carry out anything without proper insurance.-UC1: Unsafe condition due to the usage of one or more inadequate protections and guards.-UC6: Unsafe condition due to there is a danger of explosion or fire.-PF2: Personal factor due to a lack of knowledge.-WF1: Work factor due to an insufficient leadership and supervision.-WF4: Work factor due to an inadequate maintenance.-WF6: Poor labor rules.

### 4.2. Feyer and Williamson Method

#### 4.2.1. Previous Events

Three events have been identified in this accident analysis: PE1, PE2, and PE3, see Figure 5. PE1 happened before the accident, environmental event (EE), and it took place because of the coal bead methane in some irregularities in the roof of the drift. All the evidences point out that this accumulation of explosive gas was caused by:(a)The ventilation system being switched off for several hours. Subsequently, it was initiated again without following any safety protocol to dilute the methane accumulations.(b)The new exploitation method which started 23 days before the accident. The staff of the mine did not consider any possible change in the methane emission rate because of that.

The PE2 event is related to equipment (EQ), a spark was produced in the electrical installation that powered the light of the drifts a few minutes after the system was switched on due to a lack of protections against an explosive environment. This fact is attributable to:(a)An incorrect hazard level of the coal mine according to the law at the time of the accident, which was classified as a much less dangerous mine.(b)The inadequate maintenance of the electrical system.

On the other hand, PE3 is a behavior event (BE) and it was caused by a lack of knowledge of the watchman about the safety procedure needed to follow after the ventilation system had been switched off, verification of the methane levels in the drifts before the workforce enter to the mine, and the equipment is restarted. The company could not certify any type of training or safety protocol.

#### 4.2.2. Contributing Factors

Two contributing factors have been identified (Figure 5) as crucial in the occurrence of the accident.

-Unsafe procedures, CF3, attributed to the work practice (WP) on safety staff of the company. The data collected suggests that there was a lack of internal safety procedures or protocols after a long period without ventilation in a workshop or underground area. If the ventilation system had been working previously or the watchman had properly controlled the methane concentrations in the drifts, the accidents would not have happened. It is considered the most important causal factor.-Incorrect supervision (S), CF4, by the mine staff. The potential changes in the health and safety conditions were not considered after the exploitation method variation, extracting more coal per hour and, therefore, emitting more methane to the drifts. The change to a more dangerous coal mine class would have been translated to more restrictive procedures in terms of safety and according to the legislation of the time. It is considered a less important factor than the previous one.

## 5. Discussion

Both methods point out the same cause of the accident in spite of the different organization and ways of sorting it. The Feyer and Williamson classify the causes of the accident in essential events and contributing factors, both of them necessary for the chain or sequence of events in the accident. 

On the other hand, the INSST method classifies the causes in immediate causes, directly related to the accident occurrence, and basic causes, which have a strong influence in the realization of the immediate causes. These immediate causes are the equivalent to the Feyer and Williamson, most of the time, while the base causes correspond to the contributing factors.

One of the main differences found during the analysis is the number of essential events for the occurrence of the accident. The INSST can use all the events found, while the Feyer and Williamson method used only three due to that fact that it is a method that digitalizes any type of accident and then extracts global conclusions and trends. This last characteristic is very important to obtain global conclusions from the causes and workplaces where the accident took place.

Another different characteristic is the sequence of the events. The three events of the Feyer and Williamson method are temporally sorted: PE1 before the accident, PE2 before PE1, and PE3 before PE2. While the INSST the causes are sorted by levels, not by a temporal order. In each level, several groups of causes are linked to the accident or other more recent causes from the accident. Figure 4 has two grouping levels: (a) the first level is formed by three immediate causes—the accident could only happen with the combination of the three causes; (b) the second level is formed by four basic causes—the immediate causes UC1 and UC6 were developed by a combination of two basic causes, WF1 + WF4 and WF1 + WF6, respectively.

Further research should be conducted to digitalize the INSST method and create a database of the accidents analyzed. In this regard, the immediate and basic causes codes should be thoroughly defined, including much more of them. The digitalization of the method would allow deeper assessments and the implementation of better preventive measures. 

On the other hand, sometimes is difficult to discriminate between and event or a contributing factor in the Feyer and Williamson method. Besides, it can also be difficult to determine the temporary position of the events. More case studies should be analyzed using the method to avoid possible misinterpretations.

The possibility to create cause diagrams using data from accidents assessed by means of different methodologies would be another important tool to fully understand the accidents and verify the goodness of the accident analysis.

## 6. Conclusions 

The main causes of this coal mine accident have been found by means of two contrasted methods, the Feyer and Williamson method and the INSST method, suggesting that the preventive structure and organization was the root of the accident. This conclusion was reached after gathering all the information available and applying an analysis procedure that did not exist at the moment of the accident. 

One of the main advantages of using a diagram analysis, either the Feyer and Williamson or the INSST, is that it gives a better picture of the accident events, providing a very useful tool to fully understand it.

Despite the current legislation, technology, and working conditions being very different from 1975, the mistakes that took place have a similar pattern when it is compared with other fatalities nowadays. Besides, organizational failures that lead to inadequate equipment, working standards, or workplace conditions are still found as important elements in accidents. This knowledge provides a valuable information in order to help the mining industry to prevent future accidents.

## Figures and Tables

**Figure 1 ijerph-16-03615-f001:**
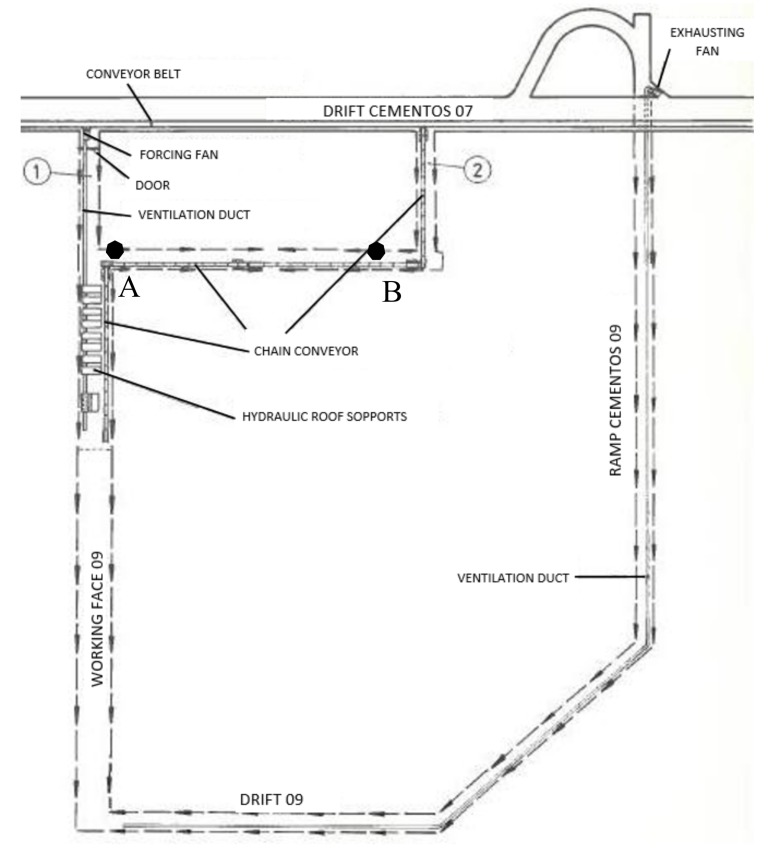
Workshop where the accident occurred. Point 1 and 2 are entry and exist of the working face, respectively, while point A and B are the measuring points.

**Figure 2 ijerph-16-03615-f002:**
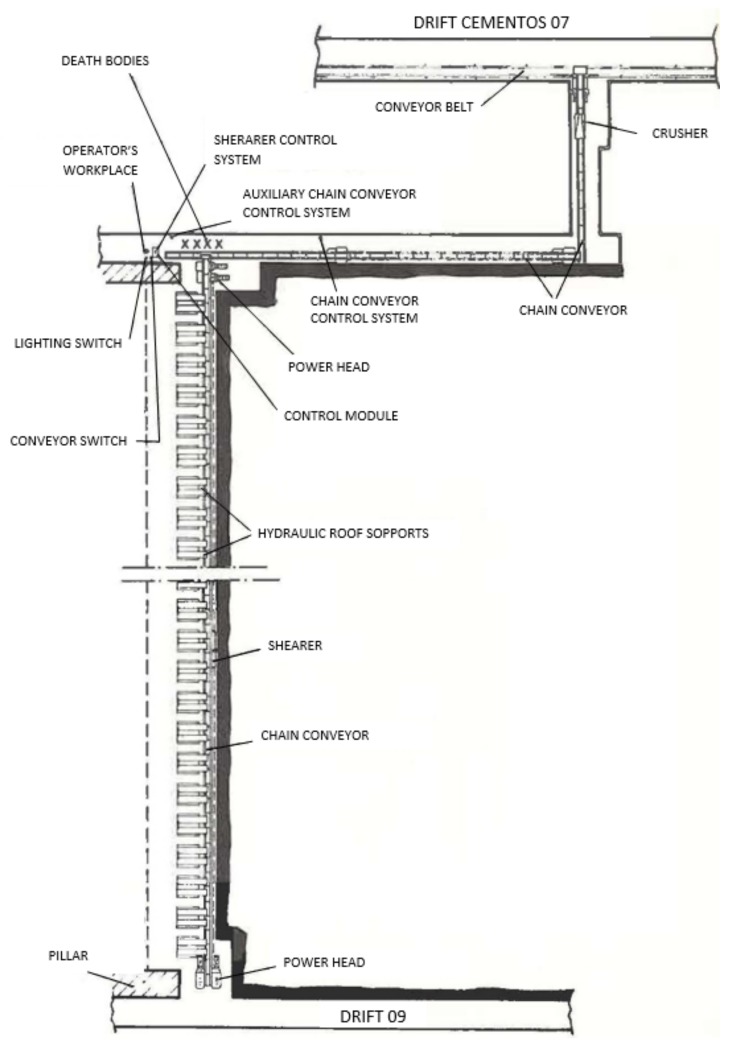
Detailed scheme of the workshop.

**Figure 3 ijerph-16-03615-f003:**
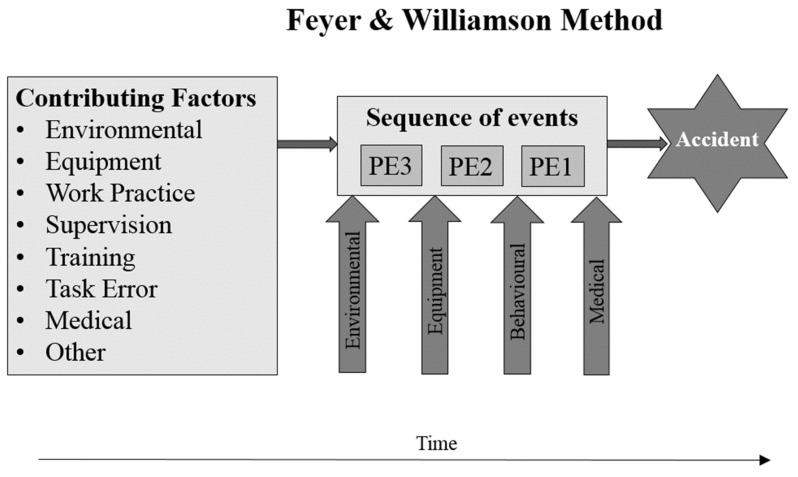
Elements that compose the Feyer and Williamson method. PE1, PE2, and PE3 are Precursor Events.

**Figure 4 ijerph-16-03615-f004:**
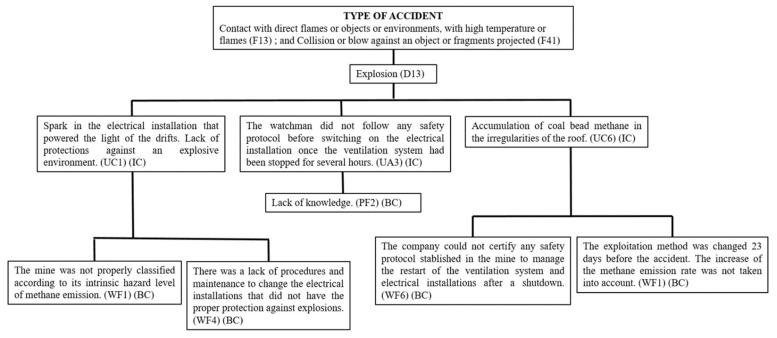
Causal tree using the Spanish Instituto Nacional de Seguridad y Salud en el Trabajo (INSST) method.

**Figure 5 ijerph-16-03615-f005:**
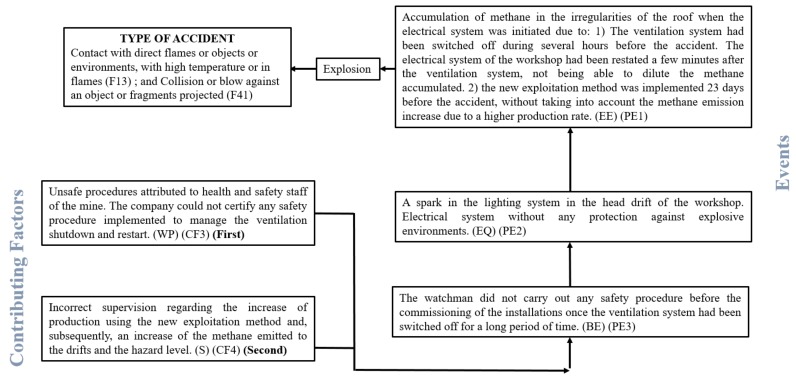
Causal tree using the Feyer and Williamson method.

**Table 1 ijerph-16-03615-t001:** Methane proportion (%).

Time	Ventilation	Point A	Point B
07:30	Switched on	0.0	0.0
07:40	Switched off	0.0	0.0
07:45	Switched off	0.2	0.0
08:00	Switched off	0.35	0.0
08:15	Switched off	0.4	0.2
08:30	Switched off	0.5	0.2
08:50	Switched off	0.7	0.7
09:00	Switched off	0.8	1.5
09:15	Switched off	0.8	3.5
09:40	Switched off	1.0	Higher than 5
09:45	Switched on	1.0	Higher than 5
09:47	Switched on	1.0	0.2
09:48	Switched on	1.0	0.0

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
