# Peer review of "Analysis of a Historical Accident in a Spanish Coal Mine"

_ijerph, 2019, doi:10.3390/ijerph16193615_

Round 1
Reviewer 1 Report
This is an interesting paper that tries to analyse the great accident occurred in a Spanish coal mine in 1975, using two well-known alternatives: 1) the method from the Spanish Instituto Nacional de Seguridad y Salud en el Trabajo (INSST), and 2) The Feyer and Williamson method.
In my opinion is very interesting the comparative between the two methods. There are several studies where is performed an analysis of an accident by some other methods. However, the analysis from two methods and their comparative is not usual.
Although the paper is interesting, I have some concerns that should be solved before possible publication.
- Please improve the accident method's description used in the study.
- Literature review is weak. Please add references about the two methods in order the readers can consult more details of them.
- Please appropriately address limitations, and future directions of research.
- The manuscript needs editing in English language. It has serious language mistakes that undermine the quality of the paper.
Author Response
The revision of the article titled Analysis of a historical accident in a Spanish coal mine has been attached, as well as the comments of the changed done in the paper below.
Please improve the accident method's description used in the study.
Further explanation has been included in this regard: lines 112-120; 134-137; 149-150
Literature review is weak. Please add references about the two methods in order the readers can consult more details of them.
Several references have been added to the manuscript
Please appropriately address limitations, and future directions of research.
Further explanation has been included in this regard: lines 267-276
The manuscript needs editing in English language. It has serious language mistakes that undermine the quality of the paper.
The paper has been thoroughly revised
Reviewer 2 Report
Overall nice paper.
Some generic comments apart from the ones in the annotated paper.
Proofreading is required. Include some international examples to add more value to the paper.Additional comments in the annotated copy.

Author Response
The revision of the article titled Analysis of a historical accident in a Spanish coal mine has been attached, as well as the comments of the changed done in the paper below.
Proofreading is required.
The paper has been thoroughly revised
Include some international examples to add more value to the paper.
Some international examples have been added in this regard
Additional comments in the annotated copy
All the changes detailed in the pdf file have been done (highlighted in green).
Comments from the ones in the annotated paper.
Some international examples have been included as mentioned. The Figures have been modified as demanded. All the changes in the text indicated by the reviewer have been done (highlighted in green).